# Factors associated with barriers to healthcare access among ever-married women of reproductive age in Bangladesh: Analysis from the 2017–2018 Bangladesh Demographic and Health Survey

**Hitomi Hinata[1]\*, Kaung Suu Lwin[1], Akifumi Eguchi[2], Cyrus Ghaznavi[3,4], Masahiro Hashizume[1], Shuhei Nomura[1,3,5]**

1 Department of Global Health Policy, Graduate School of Medicine, The University of Tokyo, Tokyo, Japan, 2 Center for Preventive Medical Sciences, Chiba University, Chiba, Japan, 3 Department of Health Policy and Management, School of Medicine, Keio University, Tokyo, Japan, 4 Department of Medicine, University of California San Francisco, San Francisco, California, United States of America, 5 Tokyo Foundation for Policy Research, Tokyo, Japan

\* tdytdc2896@gmail.com

**Data Availability Statement:** Data will be freely available after a reasonable request from the DHS

## Abstract

### Background

Globally, women experience healthcare inequalities, which may contribute to excessive mortality rates at various stages of their lives. Though Bangladesh has achieved excellent progress in providing healthcare, the country still has some critical challenges that need immediate attention. The objective of this study is to examine the association between social determinants and barriers to accessing healthcare among ever-married women aged 15–49 in Bangladesh.

### Methods

The study was conducted among 20,127 women aged 15–49, using data from the 2017–2018 Bangladesh Demographic and Health Survey. Four barriers to healthcare were considered: whether women face problems with permission, obtaining money, distance, and companionship. The multivariable logistic regression analysis was used, with a broad array of independent variables (such as age, and educational level) to identify the determinants of barriers to healthcare access. The associations were expressed as adjusted odds ratios (AOR) with a 95% confidence interval (CI).

### Results

More than two-thirds (66.3%) of women reported having at least one perceived barrier to accessing healthcare. Women with a higher level of education (AOR = 0.49, 95% CI: 0.41–0.57), owning a mobile telephone (AOR = 0.78, 95% CI: 0.73–0.84), and those in the richest wealth quintile (AOR = 0.45, 95% CI: 0.38–0.52) had lower odds of having barriers to accessing healthcare. In addition, widowed (AOR = 1.53, 95% CI: 1.26–1.84), divorced

program (ttps://dhsprogram.com/data/available-datasets.cfm), as it is not currently possible to access the data without making a request.

**Funding:** This work was partly funded by a research grant from the Ministry of Education, Culture, Sports, Science and Technology of Japan (21H03203).

**Competing interests:** No. The authors have declared that no competing interests exist.

(AOR = 1.91, 95% CI:1.47–2.48), or separated (AOR = 1.98, 95% CI: 1.46–2.69) women had higher odds of having a money barrier to accessing healthcare, than married women.

## Conclusions

This study shows that individual-, household-, and community-level factors are associated with barriers to healthcare accessibility. To improve the state of women's health in Bangladesh, it is vital to consider these socio-economic factors and implement fundamental measures, such as supporting the national health policy, empowering women's socio-economic situation, and spreading the flexible way of healthcare access.

## Introduction

Globally, women experience healthcare inequalities, which may contribute to excessive mortality rates at various stages of their lives [1]. Reports have highlighted that, in addition to education levels and poverty, numerous social, cultural, and geographical factors are associated with poor utilization and access to healthcare services [2–4]. Furthermore, previous research has indicated that individual and household factors, such as marital status [5], educational attainment [6], and wealth index [7], may be linked to women's access to healthcare services. For instance, despite the majority of maternal deaths being considered preventable, approximately 295,000 women worldwide died during pregnancy or within the postpartum period in 2017 [8].

On a global scale, Bangladesh is among the countries that have made significant progress in reducing maternal and child mortality rates [9]. Maternal mortality has decreased from 297 deaths per 100,000 live births in 2007 to 173 deaths per 100,000 live births in 2017 [8], and under-5 mortality has decreased from 49 deaths per 1,000 live births in 2010 to 29 deaths per 1,000 live births in 2020 [10]. The success of Bangladesh's "Maternal Health Strategy 2001" and its subsequent successor, the "Bangladesh National Strategy on Maternal Health (BNSMH) 2017–2030," likely underlie this progress. These strategies aim to address existing disparities and inequities in the provision of quality maternal healthcare services and to tackle the social and developmental factors that affect maternal health [9]. Despite the remarkable progress in healthcare, particularly in maternal and child health outcomes, Bangladesh is still considered to be in the process of improving healthcare access, especially ensuring access to primary and emergency healthcare services for all. Various discussions have been held regarding barriers to access, such as inadequate financial resources [11], a shortage of skilled personnel [12], and economic disparities [11], but scientific evidence regarding the scale and factors contributing to these access barriers is extremely limited.

The objective of this study is to examine the determinants and barriers to healthcare access among women of reproductive age in Bangladesh, using nationally representative data. The insights gained from this study are expected to be useful for healthcare policymakers in achieving healthcare equity, improving women's healthcare through the redistribution of health services in Bangladesh, and further reducing maternal and child mortality rates.

## Materials and methods

### Data sources

The data used for this study was obtained from the 2017–2018 Bangladesh Demographic Health Survey (BDHS) [13], which was collected cross-sectionally from October 2017 to

March 2018. The BDHS is a nationally representative survey that aims to collect data on many health-related topics, including healthcare access, demographic data, and the health status of women and children. Data will be freely available after a reasonable request from the DHS program. The survey protocol was approved by institutional review boards (IRBs) at ICF and the Bangladesh Medical Research Council (BMRC). Both IRBs and BMRC approved the protocols before the commencement of data collection activities.

## Ethical statement

In this study, as we only used the anonymous, public-use secondary data sets provided by BDHS, there was no need for ethical approval, and consequently, we did not obtain informed consent. The study was conducted following the guidelines of the Helsinki Declaration.

## Setting and population

The BDHS used a list of enumeration areas (EAs) from the 2011 Population and Housing Census of the People's Republic of Bangladesh, provided by the Bangladesh Bureau of Statistics (BBS), as a sampling frame (BBS 2011). The primary sampling unit (PSU) of the survey was an EA with an average of approximately 120 households. The survey was conducted using two-stage stratified cluster sampling. In the first stage, 675 EAs divided into 250 urban areas and 425 rural areas were selected with probability proportional to EA size. Furthermore, a complete household listing operation was performed in all selected EAs to provide a sampling frame. In the second stage, a systematic sample of an average of 30 households per EA was selected for urban and rural areas separately, and for each of the eight divisions. Based on this design, 20,250 residential households were selected. Among the 20,376 ever-married women aged 15–49 eligible for interviews, 20,127 were interviewed, yielding a response rate of 98.8%. The detail of the methodology is available in the BDHS 2017–2018 report [13].

## Variables

**Outcome variable.** The outcome variable in this study was barriers to healthcare access. This BDHS has four questions that mentioned barriers to healthcare access: including receiving permission for seeking care or treatment advice or treatment (permission), financial resources which were mentioned as obtaining money (money), distance to health facility (distance), and not wanting to present to healthcare alone (alone) and each woman was interviewed on these barriers. A woman was considered to have any given barrier in healthcare access and coded as "1" if she faced problems related to the aforementioned four kinds of each barriers. On the contrary, she was considered to have no barrier to healthcare access and coded as "0" if she did not report any barriers. Moreover, we generated a new outcome variable for having at least one barrier to healthcare access (at least one barrier). A woman was considered to have at least one barrier to healthcare access and coded as "1" if she faced any one barrier from the aforementioned four kinds of barriers.

**Independent variable.** Some previous studies indicated that both individual and community-level factors had a statistically significant association with barriers in healthcare access [5, 6, 14]. Therefore, the individual-level factors we considered in this study were based on the previous literature as well as data availability. Individual-level factors included age (15–19, 20–24, 25–29, 30–34, 35–39, 40–44, and 45–49), marital status (married, widowed, divorced, and no longer living together or separated), education level (no education, primary, secondary, and higher), employment (not working, professional or technical or managerial, sales, agricultural, household and domestic, services, and manual), religion (Islam, Hinduism, and Buddhism Christianity), health insurance status, frequency (not at all, less than once a week, and at

least once a week) of exposure to mass media (radio, newspaper or magazine, and television), the sex of the household head, and owning a mobile phone.The household-level factor consisted of household wealth status (poorest, poorer, middle, richer, and richest). Households are given scores based on the number and kinds of consumer goods they own, ranging from a television to a bicycle or car, and housing characteristics such as a source of drinking water etc., and these scores are derived using principal component analysis [13]. Household wealth quintiles are compiled by assigning the household score to each usual (de jure) household member, ranking each person in the household population by her or his score, and then dividing the distribution into five equal categories, each comprising 20% of the population [13]. The community-level factors consisted of residence (urban and rural), and region (Dhaka, Barisal, Chittagong, Khulna, Mymensingh, Rajshahi, Rangpur, Sylhet, and not dejure resident). Note that all these independent variables were categorical variables.

**Statistical analysis.** STATA/MP version 17.0 was used for analysis. The data were weighted using sampling weights, PSU, and strata. As the first step, we performed a descriptive analysis by crosstabulation of all independent variables against each four barriers and having at least one barrier. The second step was a bivariate analysis using a chi-squared test to examine the relationship of each independent variable with the outcome variables. Thirdly, we assessed the multi-collinearity issue by verifying the correlation coefficients. Referring to the assessment, we included all the independent variables in the logistic regression model. Finally, a multi-level mixed-effects logistic regression with fixed-effects of the individual-, household-, and community-level factors and random-intercept of between-cluster characteristics was constructed to examine the association between determinants and barriers in accessing healthcare at the individual-, household-, and community-level factors. A cluster is a geographic area containing approximately 50–150 households, generally based on a country's census EA. A p-value of <0.05 was considered statistically significant.

## Results

### Socio-demographic characteristics of the study participants

Table 1 presents the socio-demographic characteristics of the participants. Approximately half of the participants (45.8%) were under 30 years old, and the majority were married (94.3%). The majority had completed secondary school (87.5%), and agriculture was the most common occupation (32.7%), while nearly half were unemployed (49.8%). Few participants reported regularly listening to the radio or reading newspapers (with 95.2% and 90.7% respectively stating they did so 'not at all'). However, over half (55.0%) stated they watched television at least once a week. The prevalence of mobile phone ownership was 60.2%, and 71.5% of respondents resided in urban areas.

Perceived barriers to healthcare access among ever-married women of reproductive age in Bangladesh

Table 2 presents the background characteristics of barriers to healthcare access among ever-married women of reproductive age in Bangladesh.

More than two-thirds (66.3%) of women reported having at least one perceived barrier to healthcare access (Fig 1). About 44.4% of women mentioned that not wanting to go alone was a barrier to healthcare access. Procuring money needed for treatment (43.8%) and distance to health facilities (40.5%) were also commonly reported barriers, and 10.8% indicated that they required permission before accessing healthcare.

Over 60% of the women had at least one barrier to healthcare access in all age categories (62.6–71.4%). By education, the percentage of women who indicated they had faced at least one barrier to healthcare was highest in the no education group (77.4%) and lowest in the

**Table 1. Socio-demographic and characteristics of ever-married women aged 15–49 years in Bangladesh, 2017–18 (n = 20,127).**

| Characteristics | Weighted Frequency | Weighted Percentage |
|---|---|---|
| **Age** | | |
| 15–19 | 2063 | 10.3 |
| 20–24 | 3556 | 17.7 |
| 25–29 | 3579 | 17.8 |
| 30–34 | 3470 | 17.2 |
| 35–39 | 2879 | 14.3 |
| 40–44 | 2297 | 11.4 |
| 45–49 | 2285 | 11.4 |
| **Marital status** | | |
| Married | 18985 | 94.3 |
| Widowed | 615 | 3.1 |
| Divorced | 307 | 1.5 |
| No longer living together /separated | 222 | 1.1 |
| **Educational level** | | |
| No education | 3334 | 16.6 |
| Primary | 6290 | 31.3 |
| Secondary | 7974 | 39.6 |
| Higher | 2531 | 12.6 |
| **Occupation** | | |
| Not working | 10029 | 49.8 |
| Professional/technical /managerial | 379 | 1.9 |
| Sales | 313 | 1.6 |
| Agricultural | 6581 | 32.7 |
| Household and domestic | 365 | 1.8 |
| Services | 870 | 4.3 |
| Manual | 1588 | 7.9 |
| **Religion** | | |
| Islam | 18251 | 90.7 |
| Hinduism | 1727 | 8.6 |
| Other | 151 | 0.8 |
| **Covered by health insurance** | | |
| No | 20092 | 99.8 |
| Yes | 36 | 0.2 |
| **Frequency of listening to radio** | | |
| Not at all | 19151 | 95.2 |
| Less than once a week | 561 | 2.8 |
| At least once a week | 415 | 2.1 |
| **Frequency of reading newspaper or magazine** | | |
| Not at all | 18262 | 90.7 |
| Less than once a week | 1218 | 6.1 |
| At least once a week | 649 | 3.2 |
| **Frequency of watching television** | | |
| Not at all | 7224 | 35.9 |
| Less than once a week | 1842 | 9.2 |
| At least once a week | 11062 | 55.0 |
| **Sex of household head** | | |

*(Continued)*

**Table 1.** (Continued)

| Characteristics | Weighted Frequency | Weighted Percentage |
|---|---|---|
| Male | 17167 | 85.3 |
| Female | 2961 | 14.7 |
| **Owns a mobile telephone** | | |
| No | 8019 | 39.8 |
| Yes | 12109 | 60.2 |
| **Wealth index** | | |
| Poorest | 3744 | 18.6 |
| Poorer | 3957 | 19.7 |
| Middle | 4060 | 20.2 |
| Richer | 4184 | 20.8 |
| Richest | 4184 | 20.8 |
| **Residence** | | |
| Urban | 5729 | 28.5 |
| Rural | 14399 | 71.5 |
| **Region** | | |
| Dhaka | 4772 | 23.7 |
| Barisal | 1028 | 5.1 |
| Chittagong | 3221 | 16.0 |
| Khulna | 2175 | 10.8 |
| Mymensingh | 1416 | 7.0 |
| Rajshahi | 2588 | 12.9 |
| Rangpur | 2213 | 11.0 |
| Sylhet | 1087 | 5.4 |
| Not dejure resident | 1632 | 8.1 |

For women's individual sampling weight, the household sampling weight was multiplied by the inverse of women's individual response rate by stratum.

highest education group (46.0%). Similarly, there is a large gap between the percentage of the women who mentioned that the barrier to getting money needed for treatment in the no education (61.4%) group and that in the higher education (19.1%) group.

Regarding the frequency of reading a newspaper or magazine, 36.7% of women who read a newspaper or magazine for at least one week have at least one barrier, whereas 69.2% of the women who read that not at all have at least one barrier. For all healthcare access barriers including having at least one barrier, the percentage of women who faced any barriers in not owning a mobile phone group is higher than that in owning a mobile phone group. 80.1% of the women whose wealth index is the poorest faced at least one barrier to healthcare, while 49.9% of the women whose wealth index is the richest faced at least one barrier.

Individual- and contextual-level factors associated with barriers in healthcare access among women in Bangladesh

Table 3 presents the results of the multilevel multivariable logistic regression analysis.

The results showed that women aged 15–19 had higher odds of having barriers in receiving permission for seeking care or treatment advice or treatment (permission-related barriers) with adjusted odds ratios (AOR) of 1.34 (95% confidence interval (CI): 1.08–1.66) and not wanting to present to healthcare alone (AOR 1.54 [1.34–1.78]) compared to women aged 40–49. In contrast, women aged 15–19 had lower odds of having barriers in financial resources which were mentioned as obtaining money (financial barriers) with AOR of 0.69 [0.60–0.81])

**Table 2. Background characteristics of barriers to healthcare access among ever-married women aged 15–49 in Bangladesh, 2017–18 (n = 20,127).**

| Characteristics | Permission | | Money | | Distance | | Alone | | At least one barrier | |
|---|---|---|---|---|---|---|---|---|---|---|
| | n | % | n | % | n | % | n | % | n | % |
| **Age** | *** | | *** | | * | | *** | | *** | |
| 15–19 | 320 | 15.5 | 737 | 35.7 | 833 | 40.4 | 1098 | 53.2 | 1416 | 68.6 |
| 20–24 | 414 | 11.6 | 1320 | 37.1 | 1402 | 39.4 | 1596 | 44.9 | 2226 | 62.6 |
| 25–29 | 424 | 11.8 | 1505 | 42.1 | 1446 | 40.4 | 1554 | 43.4 | 2339 | 65.4 |
| 30–34 | 364 | 10.5 | 1591 | 45.9 | 1473 | 42.4 | 1479 | 42.6 | 2308 | 66.5 |
| 35–39 | 333 | 11.6 | 1393 | 48.4 | 1227 | 42.6 | 1237 | 43.0 | 1964 | 68.2 |
| 40–44 | 230 | 10.0 | 1115 | 48.5 | 955 | 41.6 | 1029 | 44.8 | 1583 | 68.9 |
| 45–49 | 297 | 13.0 | 1160 | 50.8 | 983 | 43.0 | 1065 | 46.6 | 1631 | 71.4 |
| **Marital status** | | | *** | | | | | | ** | |
| Married | 2229 | 11.7 | 8146 | 42.9 | 7829 | 41.2 | 8583 | 44.7 | 12649 | 66.6 |
| Widowed | 77 | 12.5 | 360 | 58.5 | 279 | 45.4 | 264 | 42.9 | 442 | 71.9 |
| Divorced | 40 | 13.0 | 177 | 57.7 | 126 | 41.0 | 122 | 39.7 | 211 | 68.7 |
| No longer living together/separated | 33 | 14.9 | 137 | 61.7 | 84 | 37.8 | 86 | 38.7 | 163 | 73.4 |
| **Educational level** | *** | | *** | | *** | | *** | | *** | |
| No education | 466 | 14.0 | 2047 | 61.4 | 1656 | 49.7 | 1648 | 49.4 | 2580 | 77.4 |
| Primary | 804 | 12.8 | 3315 | 52.7 | 2827 | 44.9 | 3016 | 47.9 | 4607 | 73.2 |
| Secondary | 937 | 11.8 | 2976 | 37.3 | 3123 | 39.2 | 3571 | 44.8 | 5114 | 64.1 |
| Higher | 172 | 6.8 | 483 | 19.1 | 712 | 28.1 | 821 | 32.4 | 1164 | 46.0 |
| **Occupation** | *** | | *** | | *** | | *** | | *** | |
| Not working | 1347 | 13.4 | 3805 | 37.9 | 3901 | 38.9 | 4459 | 44.5 | 6328 | 63.1 |
| Professional /technical /managerial | 21 | 5.5 | 51 | 13.5 | 88 | 23.2 | 85 | 22.4 | 141 | 37.2 |
| Sales | 38 | 12.1 | 136 | 43.5 | 119 | 38.0 | 111 | 35.5 | 194 | 62.0 |
| Agricultural | 719 | 10.9 | 3422 | 52.0 | 3152 | 47.9 | 3264 | 49.6 | 4922 | 74.8 |
| Household and domestic services | 41 | 11.2 | 254 | 69.6 | 156 | 42.7 | 156 | 42.7 | 272 | 74.5 |
| Services | 84 | 9.7 | 418 | 48.0 | 310 | 35.6 | 334 | 38.4 | 552 | 63.4 |
| Manual | 131 | 8.2 | 735 | 46.3 | 592 | 37.3 | 644 | 40.6 | 1054 | 66.4 |
| **Religion** | | | *** | | ** | | * | | ** | |
| Islam | 2171 | 11.9 | 7906 | 43.3 | 7448 | 40.8 | 8133 | 44.6 | 12122 | 66.4 |
| Hinduism | 192 | 11.1 | 820 | 47.5 | 786 | 45.5 | 843 | 48.8 | 1223 | 70.8 |
| Other (Buddhism & Christianity) | 16 | 10.6 | 94 | 62.3 | 82 | 54.3 | 79 | 52.3 | 120 | 79.5 |
| **Covered by health insurance** | | | | | | | | | | |
| No | 2373 | 11.8 | 8803 | 43.8 | 8298 | 41.3 | 9033 | 45.0 | 13441 | 66.9 |
| Yes | 6 | 16.7 | 16 | 44.4 | 18 | 50.0 | 22 | 61.1 | 23 | 63.9 |
| **Frequency of listening to radio** | | | *** | | *** | | *** | | *** | |
| Not at all | 2279 | 11.9 | 8558 | 44.7 | 8009 | 41.8 | 8670 | 45.3 | 12929 | 67.5 |
| Less than once a week | 63 | 11.2 | 150 | 26.7 | 183 | 32.6 | 229 | 40.8 | 308 | 54.9 |
| At least once a week | 38 | 9.2 | 111 | 26.7 | 124 | 29.9 | 155 | 37.3 | 225 | 54.2 |
| **Frequency of reading newspaper or magazine** | *** | | *** | | *** | | *** | | *** | |
| Not at all | 2247 | 12.3 | 8453 | 46.3 | 7818 | 42.8 | 8493 | 46.5 | 12645 | 69.2 |
| Less than once a week | 98 | 8.0 | 263 | 21.6 | 344 | 28.2 | 406 | 33.3 | 582 | 47.8 |
| At least once a week | 34 | 5.2 | 104 | 16.0 | 153 | 23.6 | 156 | 24.0 | 238 | 36.7 |
| **Frequency of watching television** | *** | | *** | | *** | | *** | | *** | |
| Not at all | 938 | 13.0 | 3903 | 54.0 | 3578 | 49.5 | 3708 | 51.3 | 5415 | 75.0 |
| Less than once a week | 284 | 15.4 | 915 | 49.7 | 782 | 42.5 | 813 | 44.1 | 1319 | 71.6 |
| At least once a week | 1158 | 10.5 | 4001 | 36.2 | 3956 | 35.8 | 4534 | 41.0 | 6731 | 60.8 |
| **Owns a mobile telephone** | *** | | *** | | *** | | *** | | *** | |

*(Continued)*

**Table 2.** (Continued)

| Characteristics | Permission | | Money | | Distance | | Alone | | At least one barrier | |
|---|---|---|---|---|---|---|---|---|---|---|
| | n | % | n | % | n | % | n | % | n | % |
| No | 1113 | 13.9 | 4269 | 53.2 | 3687 | 46.0 | 4012 | 50.0 | 5973 | 74.5 |
| Yes | 1265 | 10.4 | 4550 | 37.6 | 4629 | 38.2 | 5042 | 41.6 | 7491 | 61.9 |
| **Sex of household head** | | | | | | | ** | | | |
| Male | 2031 | 11.8 | 7530 | 43.9 | 7056 | 41.1 | 7792 | 45.4 | 11522 | 67.1 |
| Female | 348 | 11.8 | 1289 | 43.5 | 1259 | 42.5 | 1262 | 42.6 | 1942 | 65.6 |
| **Wealth index** | *** | | *** | | *** | | *** | | *** | |
| Poorest | 533 | 14.2 | 2443 | 65.3 | 2039 | 54.5 | 1944 | 51.9 | 2998 | 80.1 |
| Poorer | 495 | 12.5 | 2151 | 54.4 | 1874 | 47.4 | 1938 | 49.0 | 2988 | 75.5 |
| Middle | 547 | 13.5 | 1739 | 42.8 | 1716 | 42.3 | 1900 | 46.8 | 2762 | 68.0 |
| Richer | 480 | 11.5 | 1456 | 34.8 | 1531 | 36.6 | 1849 | 44.2 | 2631 | 62.9 |
| Richest | 326 | 7.8 | 1031 | 24.6 | 1156 | 27.6 | 1424 | 34.0 | 2087 | 49.9 |
| **Residence** | *** | | *** | | *** | | *** | | *** | |
| Urban | 508 | 8.9 | 2163 | 37.8 | 1791 | 31.3 | 2137 | 37.3 | 3318 | 57.9 |
| Rural | 1871 | 13.0 | 6656 | 46.2 | 6525 | 45.3 | 6917 | 48.0 | 10146 | 70.5 |
| **Region** | *** | | *** | | *** | | *** | | *** | |
| Dhaka | 622 | 13.0 | 2027 | 42.5 | 1946 | 40.8 | 2107 | 44.2 | 3139 | 65.8 |
| Barisal | 96 | 9.3 | 482 | 46.9 | 507 | 49.3 | 509 | 49.5 | 739 | 71.9 |
| Chittagong | 403 | 12.5 | 1307 | 40.6 | 1241 | 38.5 | 1380 | 42.8 | 2022 | 62.8 |
| Khulna | 224 | 10.3 | 1024 | 47.1 | 975 | 44.8 | 1079 | 49.6 | 1578 | 72.6 |
| Mymensingh | 109 | 7.7 | 730 | 51.6 | 596 | 42.1 | 638 | 45.1 | 975 | 68.9 |
| Rajshahi | 267 | 10.3 | 1011 | 39.1 | 934 | 36.1 | 1055 | 40.8 | 1660 | 64.1 |
| Rangpur | 324 | 14.6 | 1175 | 53.1 | 1014 | 45.8 | 1055 | 47.7 | 1647 | 74.4 |
| Sylhet | 134 | 12.3 | 500 | 46.0 | 501 | 46.1 | 524 | 48.2 | 747 | 68.7 |
| Not dejure resident | 203 | 12.4 | 567 | 34.7 | 605 | 37.1 | 711 | 43.6 | 958 | 58.7 |

*p<0.05

**p<0.01

***p<0.001

compared to women aged 40–49. Compared to married women, widowed, divorced, and separated women had higher odds of financial barriers (AOR 1.53 [1.26–1.84], 1.91 [1.47–2.48], and 1.98 [1.46–2.69]), respectively. Women with higher education had lower odds of all barriers which are permission-related barriers (0.61 [0.47–0.78]), financial barriers (AOR 0.39 [0.33–0.46]), barriers of distance to health facility (distance-related barriers, AOR 0.61 [0.52–0.71]), and having at least one barrier to healthcare access (AOR 0.49 [0.41–0.57]). Higher education was associated with fewer barriers in a dose-dependent manner. With respect to occupational status, women engaged in household or domestic services had higher odds of financial barriers (AOR of 2.07 [1.59–2.69]) compared to the women who did not work. In contrast, women engaged in professional, technical, or managerial work had lower odds of having financial barriers (AOR 0.57 [0.40–0.80]), having distance-related barriers (AOR 0.75 [0.57–0.99]), not wanting to present to healthcare alone (AOR 0.65 [0.49–0.85]), and having at least one barrier (AOR 0.71 [0.55–0.91]) compared to the women who did not work.

With respect to the use of information sources, women who read newspapers or magazines at least once a week had lower odds of having financial barriers (AOR 0.67 [0.52–0.86]), distance-related barriers (AOR 0.83 [0.72–0.96]), not wanting to present to healthcare alone (AOR 0.67 [0.54–0.83]), and having at least one barrier (AOR 0.65 [0.53–0.79]) than those

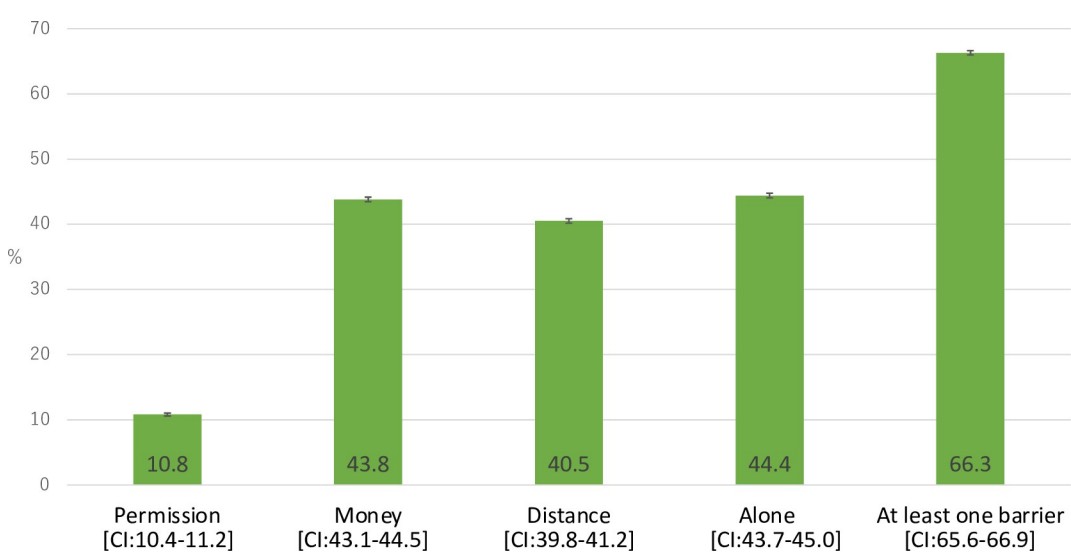

**Fig 1. The proportion of barriers to access to healthcare among women aged 15–49 in Bangladesh.** Permission: receiving permission for seeking care or treatment advice or treatment, Money: financial resources which were mentioned as obtaining money, Distance: distance to health facility, Alone: not wanting to present to healthcare alone, CI: 95% confidence interval.

who did not read them at all. Similarly, women who read newspapers or magazines less than once a week also had lower odds of having the aforementioned barriers without distance-related barriers than those who did not read them at all. In addition, women who watched television at least once a week had lower odds of having permission-related barriers (AOR 0.87 [0.77–0.99]), distance-related barriers (AOR 0.89 [0.82–0.97]), not wanting to present to healthcare alone (AOR 0.83 [0.77–0.90]) and having at least one barrier (AOR 0.88 [0.80–0.96]). Compared to those who did not watch television at all. Women who had a mobile phone had lower odds of having permission-related barriers (AOR 0.81 [0.73–0.90]), financial barriers (AOR 0.79 [0.73–0.84]), distance-related barriers (AOR 0.91 [0.85–0.97]), not wanting to present to healthcare alone (AOR 0.85 [0.79–0.91]) and having at least one barrier (AOR 0.78 [0.73–0.84]) than women who did not have a mobile phone.

Compared to women in the lowest wealth quantile, women in the highest wealth quantile had lower odds of having financial barriers (AOR 0.20 [0.17–0.24]), distance-related barriers (AOR 0.55 [0.48–0.64]) and having at least one barrier (AOR 0.45 [0.38–0.52]). For all aforementioned barriers, we observed a dose-dependent relationship between higher wealth and fewer barriers. Women who reside in rural areas have higher odds of having permission-related barriers (AOR 1.32 [1.02–1.72]), distance-related barriers (AOR 1.49 [1.29–1.72]), not wanting to present to healthcare alone (AOR 1.27 [1.12–1.44]) and having at least one barrier (AOR 1.18 [1.02–1.36]) compared to women who reside in urban areas. Alternatively, we found that women who reside in rural areas have higher odds of financial barriers (AOR 0.77 [0.67–0.89]). Regarding regional variation, compared to those residing in Dhaka, those in Chittagong (AOR 0.78 [0.64–0.95]), Mymensingh (AOR 0.78 [0.61–0.99]), Rajshahi (AOR 0.70 [0.56–0.86]), and Sylhet (AOR 0.74 [0.58–0.96]) had lower odds of having at least one barrier.

## Discussion

This study is the first comprehensive analysis that examined the associations between individual-, household-, and community-level factors and barriers in accessing healthcare among

**Table 3. Multivariable logistic regression of individual and community level factors associated with barriers to healthcare access among ever-married women aged 15–49 in Bangladesh, 2017–18 (n = 20,127).**

| Characteristics | Permission | Money | Distance | Alone | At least one barrier |
|---|---|---|---|---|---|
| | AOR [95% CI] | AOR [95% CI] | AOR [95% CI] | AOR [95% CI] | AOR [95% CI] |
| **Age** | | | | | |
| 15–19 | 1.34 [1.08–1.66]** | 0.69 [0.60–0.81]*** | 1.06 [0.92–1.23] | 1.54 [1.34–1.78]*** | 1.17 [1.00–1.37] |
| 20–24 | 1.09 [0.90–1.33] | 0.84 [0.73–0.96]** | 1.13 [0.99–1.28] | 1.17 [1.03–1.33]* | 0.99 [0.86–1.13] |
| 25–29 | 1.05 [0.87–1.28] | 0.94 [0.83–1.07] | 1.10 [0.97–1.25] | 1.07 [0.95–1.21] | 1.02 [0.89–1.17] |
| 30–34 | 0.93 [0.77–1.12] | 1.06 [0.94–1.20] | 1.14 [1.00–1.28]* | 0.99 [0.87–1.11] | 1.01 [0.88–1.15] |
| 35–39 | 0.95 [0.79–1.14] | 1.03 [0.91–1.17] | 1.08 [0.95–1.22] | 0.95 [0.84–1.07] | 0.97 [0.85–1.11] |
| 40–44 | 0.79 [0.65–0.96]* | 0.95 [0.83–1.08] | 0.98 [0.86–1.12] | 0.98 [0.87–1.11] | 0.94 [0.82–1.08] |
| 45–49 | Ref | Ref | Ref | Ref | Ref |
| **Marital status** | | | | | |
| Married | - | Ref | - | - | Ref |
| Widowed | - | 1.53 [1.26–1.84]*** | - | - | 1.13 [0.92–1.38] |
| Divorced | - | 1.91 [1.47–2.48]*** | - | - | 1.15 [0.88–1.51] |
| No longer living together/separated | - | 1.98 [1.46–2.69]*** | - | - | 1.22 [0.88–1.69] |
| **Educational level** | | | | | |
| No education | Ref | Ref | Ref | Ref | Ref |
| Primary | 0.89 [0.77–1.03] | 0.83 [0.75–0.92]*** | 0.84 [0.76–0.93]*** | 0.94 [0.86–1.04] | 0.87 [0.78–0.97]* |
| Secondary | 0.81 [0.68–0.95]** | 0.61 [0.55–0.69]*** | 0.75 [0.68–0.84]*** | 0.84 [0.76–0.93]** | 0.68 [0.60–0.77]*** |
| Higher | 0.61 [0.47–0.78]*** | 0.39 [0.33–0.46]*** | 0.61 [0.52–0.71]*** | 0.63 [0.55–0.73]*** | 0.49 [0.41–0.57]*** |
| **Occupation** | | | | | |
| Not working | Ref | Ref | Ref | Ref | Ref |
| Professional/technical/managerial | 0.71 [0.43–1.18] | 0.57 [0.40–0.80]** | 0.75 [0.57–0.99]* | 0.65 [0.49–0.85]** | 0.71 [0.55–0.91]** |
| Sales | 1.25 [0.85–1.82] | 1.20 [0.92–1.55] | 1.08 [0.84–1.40] | 0.75 [0.58–0.96]* | 1.02 [0.79–1.32] |
| Agricultural | 0.92 [0.81–1.06] | 1.16 [1.06–1.26]** | 1.08 [0.99–1.17] | 1.03 [0.94–1.11] | 1.20 [1.09–1.31]*** |
| Household and domestic services | 0.93 [0.63–1.36] | 2.07 [1.59–2.69]*** | 1.22 [0.95–1.55] | 0.99 [0.78–1.25] | 1.31 [1.00–1.73] |
| Services | 0.85 [0.65–1.13] | 1.28 [1.08–1.52]** | 0.91 [0.77–1.08] | 0.74 [0.63–0.88]*** | 0.88 [0.74–1.05] |
| Manual | 0.76 [0.62–0.94]* | 1.26 [1.11–1.42]*** | 0.93 [0.82–1.05] | 0.88 [0.78–0.99]* | 1.08 [0.95–1.22] |
| **Religion** | | | | | |
| Islam | - | Ref | Ref | Ref | Ref |
| Hinduism | - | 1.08 [0.94–1.25] | 1.06 [0.93–1.22] | 1.16 [1.02–1.33]* | 1.11 [0.96–1.29] |
| Other (Buddhism &Christianity) | - | 1.56 [0.98–2.50] | 1.55 [0.99–2.40] | 1.40 [0.92–2.15] | 1.51 [0.91–2.52] |
| **Covered by health insurance** | | | | | |
| No | - | - | - | - | - |
| Yes | - | - | - | - | - |
| **Frequency of listening to radio** | | | | | |
| Not at all | - | Ref | Ref | Ref | Ref |
| Less than once a week | - | 0.85 [0.69–1.06] | 0.95 [0.77–1.15] | 1.12 [0.93–1.36] | 0.96 [0.79–1.17] |
| At least once a week | - | 0.83 [0.64–1.06] | 0.88 [0.70–1.11] | 0.99 [0.79–1.23] | 0.95 [0.76–1.19] |
| **Frequency of reading newspaper or magazine** | | | | | |
| Not at all | Ref | Ref | Ref | Ref | Ref |
| Less than once a week | 0.84 [0.66–1.07] | 0.69 [0.58–0.81]*** | 0.83 [0.72–0.96]* | 0.75 [0.65–0.87]*** | 0.72 [0.62–0.82]*** |
| At least once a week | 0.72 [0.49–1.08] | 0.67 [0.52–0.86]** | 0.89 [0.72–1.11] | 0.67 [0.54–0.83]*** | 0.65 [0.53–0.79]*** |
| **Frequency of watching television** | | | | | |
| Not at all | Ref | Ref | Ref | Ref | Ref |
| Less than once a week | 1.15 [0.97–1.36] | 1.03 [0.91–1.16] | 0.88 [0.78–0.98]* | 0.80 [0.71–0.89]*** | 0.98 [0.86–1.12] |
| At least once a week | 0.87 [0.77–0.99]* | 0.93 [0.85–1.01] | 0.89 [0.82–0.97]** | 0.83 [0.77–0.90]*** | 0.88 [0.80–0.96]** |

*(Continued)*

**Table 3.** (Continued)

| Characteristics | Permission | Money | Distance | Alone | At least one barrier |
|---|---|---|---|---|---|
| | AOR [95% CI] | AOR [95% CI] | AOR [95% CI] | AOR [95% CI] | AOR [95% CI] |
| **Owns a mobile telephone** | | | | | |
| No | Ref | Ref | Ref | Ref | Ref |
| Yes | 0.81 [0.73–0.90]*** | 0.79 [0.73–0.84]*** | 0.91 [0.85–0.97]** | 0.85 [0.79–0.91]*** | 0.78 [0.73–0.84]*** |
| **Sex of household head** | | | | | |
| Male | - | - | - | Ref | - |
| Female | - | - | - | 0.88 [0.81–0.96]** | - |
| **Wealth index** | | | | | |
| Poorest | Ref | Ref | Ref | Ref | Ref |
| Poorer | 0.84 [0.72–0.98]* | 0.67 [0.60–0.74]*** | 0.82 [0.74–0.91]*** | 0.97 [0.88–1.08] | 0.85 [0.75–0.96]* |
| Middle | 0.91 [0.77–1.08] | 0.45 [0.40–0.51]*** | 0.75 [0.67–0.84]*** | 1.01 [0.91–1.13] | 0.67 [0.59–0.76]*** |
| Richer | 0.84 [0.70–1.02] | 0.32 [0.28–0.36]*** | 0.68 [0.60–0.77]*** | 1.05 [0.93–1.18] | 0.60 [0.53–0.69]*** |
| Richest | 0.73 [0.59–0.92]** | 0.20 [0.17–0.24]*** | 0.55 [0.48–0.64]*** | 0.87 [0.76–1.00] | 0.45 [0.38–0.52]*** |
| **Residence** | | | | | |
| Urban | Ref | Ref | Ref | Ref | Ref |
| Rural | 1.32 [1.02–1.72]* | 0.77 [0.67–0.89]*** | 1.49 [1.29–1.72]*** | 1.27 [1.12–1.44]*** | 1.18 [1.02–1.36]* |
| **Region** | | | | | |
| Dhaka | Ref | Ref | Ref | Ref | Ref |
| Barisal | 0.75 [0.49–1.16] | 0.79 [0.62–1.01] | 1.19 [0.93–1.53] | 1.12 [0.90–1.40] | 0.92 [0.71–1.20] |
| Chittagong | 0.93 [0.67–1.29] | 0.84 [0.69–1.03] | 0.86 [0.71–1.05] | 0.93 [0.78–1.11] | 0.78 [0.64–0.95]* |
| Khulna | 0.95 [0.66–1.38] | 1.03 [0.83–1.28] | 1.03 [0.83–1.27] | 1.17 [0.97–1.42] | 1.02 [0.82–1.26] |
| Mymensingh | 0.53 [0.35–0.81]** | 0.94 [0.75–1.19] | 0.79 [0.63–1.00] | 0.90 [0.73–1.11] | 0.78 [0.61–0.99]* |
| Rajshahi | 0.92 [0.64–1.31] | 0.66 [0.53–0.81]*** | 0.75 [0.60–0.92]** | 0.83 [0.69–1.00]* | 0.70 [0.56–0.86]** |
| Rangpur | 0.97 [0.67–1.40] | 0.86 [0.69–1.07] | 0.92 [0.74–1.14] | 0.97 [0.80–1.18] | 0.90 [0.71–1.13] |
| Sylhet | 0.88 [0.58–1.35] | 0.77 [0.60–0.98]* | 0.97 [0.76–1.24] | 1.00 [0.81–1.25] | 0.74 [0.58–0.96]* |
| Not dejure resident | 1.01 [0.77–1.31] | 0.77 [0.65–0.91]** | 0.80 [0.67–0.94]** | 0.88 [0.75–1.02] | 0.64 [0.54–0.76]*** |
| AUC | 0.83 [0.82–0.83] | 0.77 [0.77–0.78] | 0.74 [0.73–0.74] | 0.71 [0.70–0.71] | 0.76 [0.75–0.76] |

*p<0.05

**p<0.01

***p<0.001

AOR: adjusted odds ratio, CI: confidence interval, Ref: reference, AUR: Area under curve

Bangladesh's ever-married women of reproductive age using BDHS data from 2017–2018. We found that two out of every three women in Bangladesh face barriers to accessing healthcare. Not wanting to present to healthcare alone, getting money needed for treatment, and distance to health facility comprised the most-commonly encountered barriers. The individual factors associated with barriers to accessing healthcare included marital status, educational level, occupation, frequency of reading newspapers or magazines, frequency of watching television, and ownership of a mobile phone. Household and community-level factors associated with barriers to healthcare included the wealth quintile, place of residence, and region. From the following, this study focused on the main variable we identified as independent variables to discuss useful suggestions for policy making.

Our study identified that the proportion of the barriers to access to healthcare in Bangladesh (66.3%) is higher than other reports in Ghana (51%, 2020) [5], Rwanda (64%, 2019) [15], and Tanzania (65%, 2018) [16]. It might be partially described by socio-cultural and economic differences among countries which may affect health-seeking behaviors [14]. Bangladesh is a

patriarchal society and gender inequality is widespread [17]. It ranks 133 of 189 countries (in 2019) in the Gender Inequality Index (GII) [18] and scores 0.54 (in 2019) in the Gender Development Index (GDI) [19]. Gender inequality in Bangladesh is characterized by limited female access to economic resources [20], male guardianship [21], control over women's life choices [22], etc. These socio-cultural practices in Bangladesh may be the factors that make women less empowered and depend on men economically for their needs [23]. These paternalistic cultural may affect the barriers to accessing healthcare including receiving permission for seeking care or treatment advice or treatment.

The marital status of women is also important for determining the odds of facing barriers in healthcare access, especially financial resources which were mentioned as obtaining money in this study. Widowed, divorced, or separated women are more likely to face barriers to healthcare access and this finding was in line with previous studies in Malaysia [24] Indonesia [25], and Tanzania [16]. The study in Ethiopia [14] using the 2016 Ethiopia DHS found that the likelihood of having perceived barriers to healthcare access among divorced or separated women was increased by 34% compared to women married or living together. These could be explained by evidence that married women may have better economic and psychosocial support from their partners to healthcare access [26]. Being married may be the contribution to better health [27], and also attributable to proper healthcare access and utilization.

The findings of our study also showed that educational level and wealth index were significant determinants of healthcare access among ever-married women of reproductive age in Bangladesh. In addition, our study established that wealth as a household-level factor also had a significant association with healthcare access. Similar findings have been reported in previous studies in Ghana [5], Afghanistan [28], WHO global health survey [29], and sub-Saharan Africa [30]. Education is a large factor in higher employment opportunities, earning an individual, household, and national economic growth [31, 32]. Educated women are more likely to engage in high-paying jobs, so they can easily afford medical expenses regardless of the cost and geographic location [33]. It may in turn promote accessibility for healthcare services [34, 35].

In our study, it was also found that exposure to mass media, especially newspapers, magazines, or television, decreased the odds of barriers to healthcare access. Women who reported never reading newspapers, magazines, or watching television each week had higher odds of facing barriers in healthcare access compared with those who were exposed to newspapers, magazines, or television. It was confirmed in previous studies in India [36], and rural Malawi [37]. The exposure to mass media which has been reported as an important determinant of healthcare utilization promotes obtaining information and health literacy, and it informs women about how to overcome barriers to accessing healthcare [38]. In contrast, the literate of adult women (aged 15-) in Bangladesh is 72.0% (in 2020, United Nations Educational, Scientific and Cultural Organization; UNESCO) [39], and BDHS reported that it was 73.3% [13]. Thus, the literacy status may be the confounding factor.

Our research also reported that women who have their own mobile telephones are less likely to face barriers to access to healthcare. A previous study showed that mobile phone use and media access are associated with the use of maternity health services [40]. Owning a mobile telephone is one of the ways to arrange transportation in an emergency [41] and access health information, engage with health practitioners, and provide quality health services. Hence, providing women with mobile access is no less crucial to safeguarding their rights than exposure to newspapers, magazines, or television to access and use health information [42]. Therefore, owning a mobile telephone contributes to women's healthcare accessibility and as a result, it causes the barriers to healthcare access among women decreased. However, owning a mobile telephone takes costs for not only the mobile telephone tariff but the price of the

phone. These costs affect the financial situation of women or households. Hence, finances may be the confounding factor.

A women's place of residence is also an important determinant of the barriers they my face in accessing healthcare. Our study indicates that women living in rural areas are less likely to face financial barriers. Bangladesh is one of the countries where increasing income disparity is of concern with a Gini coefficient of 0.458 (2010) [43]. Furthermore, the income Gini coefficients in rural and urban areas were 0.431 (2010) and 0.452 (2010), respectively [43]. This suggests that income inequality tends to be more severe in urban areas than in rural areas. Therefore, income disparity might be one factor making it harder for urban women, compared to those in rural areas, to secure the necessary financial resources for healthcare.

## Strengths and limitations of this study

The nationally representative data allowed us to comprehensively assess women's barriers in accessing healthcare. The data has a high response rate, and the study's methodology followed best practices such as gathering data with experienced data collectors and multi-stage sampling. The findings can be generalized to all women of their reproductive ages in Bangladesh. Moreover, the study employed advanced statistical models that accounted for individual-and community-level factors.

Despite all, this study has some limitations. First, the cross-sectional study design restricted our capacity to draw underlying deductions for the cause-effect relationship, which require a longitudinal design, but could not be determined. Second, due to the limited number of variables collected by the 2017–2018 BDHS, we could not examine complete factors related to healthcare accessibility, particularly the health system and health worker-related factors. Third, the characteristics of DHS questionnaire regarding some individual situations or status might be subjective to social desirability bias. For example, some of the respondents are afraid to mention the barriers that might lead to underestimation.

## Conclusion

This study shows that the individual-, household-, and community-level factors are associated with barriers to healthcare accessibility. Specifically, age, marital status, educational level, employment, exposure to mass media, owning a mobile phone, wealth status, and residence are associated with barriers to healthcare accessibility. To improve the state of women's health in Bangladesh, it is vital to consider these socio-economic factors and implement fundamental measures, such as supporting the national health policy, empowering women's socio-economic situation, and spreading the flexible way of healthcare access.

## Supporting information

**S1 Table. The proportion of barriers to access to healthcare among women aged 15–49 in Bangladesh.**
(DOCX)

## Author Contributions

**Conceptualization:** Hitomi Hinata.

**Data curation:** Hitomi Hinata.

**Formal analysis:** Hitomi Hinata.

**Investigation:** Hitomi Hinata.

**Methodology:** Hitomi Hinata, Kaung Suu Lwin, Akifumi Eguchi, Masahiro Hashizume, Shuhei Nomura.

**Project administration:** Shuhei Nomura.

**Supervision:** Masahiro Hashizume, Shuhei Nomura.

**Writing – original draft:** Hitomi Hinata.

**Writing – review & editing:** Hitomi Hinata, Kaung Suu Lwin, Akifumi Eguchi, Cyrus Ghaznavi, Masahiro Hashizume, Shuhei Nomura.

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
