## [Decision Letter · Decision Letter 0]

3 May 2023

PONE-D-23-06003Factors associated with barriers to healthcare access among ever-married women of reproductive age in Bangladesh: analysis from the 2017-2018 Bangladesh Demographic and Health SurveyPLOS ONE

Dear Dr. Hitomi,,

Thank you for submitting your manuscript to PLOS ONE. After careful consideration, we feel that it has merit but does not fully meet PLOS ONE’s publication criteria as it currently stands. Therefore, we invite you to submit a revised version of the manuscript that addresses the points raised during the review process.

Comments of Reviewer I:

1. Abstract: in background: Please specify what remarkable changes have been made by Bangladesh if possible in numeric value.

2. Abstract: Methods: It would be better to provide the independent variables and how did you analyze them? Also, provide the statistical software that you used for analysing the data. Did you manipulate the variables and data? If yes provide the information?

3. Abstract: Result: Present the result of bivariate analysis also.

4. Abstract: Conclusions: Pinpoint how the policy-makers can utilize your result.

5. Data Availability: I would recommend writing as 'Data will be freely available after the reasonable request from the DHS program'. Since it is not possible to access the data without a request.

6. Introduction: Can you present the findings of previous BDHSs? Were there any changes and which variables are remarkably changed compared to previous one? What are the possible factors for that?

7. Materials and methods: More elaboration is needed here. Follow STROBE guidelines. I am confusing the sentence: 'In the second stage, 20,2500 households were selected for data collection,….' Please revisit it.

8. Statistical analysis: Citation required here at: ..' STATA/MP version 17.0 was used for analysis.' How did you assess the variables in multivariate analysis? Did you assess the multi-collinearity issue? What is the basis of assessing the variables?

9. Statistical analysis: In the multivariate analysis, there is no consistency in assuming reference value? For example, last attribute/category in age group, while first category/attributes for others? What is the cause of that? Sometimes, it will be interesting if you make the middle attribute as reference category such as in wealth quintile.

10. Result: Present the impression of the table rather than an explanation of the table exactly. What is the basis of 5-year age category?

11. Result: … 5.7% of respondents were widowed, divorced..' Do not start the sentence with number. You may write as 'almost six (5.7) percent of the total respondents…'

In table 3: Wealth Index: The reference category in the middle would be interesting to compare with the poorest and the richest.

12. Result: In table 3: Residence: Rural woman OR = 0.77. It is interesting. Are the rural women rich in Bangladesh? ….The women who lived in rural areas had less likely to have financial barrier compared to those women who lived in urban areas...Am I right? Please present the possible causes for that.

13. Result in table 3: It is not necessary to indicate ** for significant value. The value of 95%CI shows whether it is significant or not. Present the model summary of each dependent variables. It shows the predictive capacity of the table. If possible provide value of AUR(Area Under Curve).

14. Discussion: Present your discussion theme wise i.e. Autonomy, Money, Proximity, Alone etc. Add these themes as sub heading of the discussion section.

15. Discussion: ..'Our study identified that the proportion of the barriers to access to healthcare in Bangladesh…' Present the exact value.

16. Conclusion: Please be specific while suggesting to the policy-makers. How your findings can be translated into policy-making?

17. Ethical Issue: I think, ethical approval was taken by organization while conducting BDHS.

18. Ethical approval and consent to participate: I would recommend to write as '…. The survey protocol was approved by institutional review boards (IRBs) at ICF and the Bangladesh Medical Research Council (BMRC). Both IRBs and BMRC approved the protocols before the commencement of data collection activities.. Consent was taken before administered…. '

19. Data availability: Revise your sentence. It is not possible to access the data without login and proposal. There is no data accessible through the link.

20. Others: Merge the citation as required. Provide links or DOI as possible and applicable so that readers could visit the original sources easily. Revisit the paper for grammar and sentence structure. Somewhere, there is no consistency within and among the sentences and paragraphs.

Comments of Reviewer # 2.

The authors have worked on the document and expressed their issues with proper objectives. There are no issues to publish this paper. However, there are some concerns in the paper. The transition of the introduction is not well. The author must try to write it smoothly and transition the global to the scenario of Bangladesh in a smooth fashion. Furthermore, I am not well convinced on the way why this research is required, therefore, please extend your introduction part so that you can grab the attention of the readers.

Some issues in following pages which I have noticed should be addressed.

Page no. 3, Line 10: LMICs- Low and Middle Income Countries (you have written Low and Lower and Middle Income Countries).

Page 5: Data Source- It would be wise to mention the ethical clearance issue of the data.

Page 5, Line 15: Please check the numbers.

With good editors and some transition in the paragraph, the paper will be readable. Therefore, please consider editing it.

We look forward to receiving your revised manuscript.

Kind regards,

Ramesh Adhikari

Academic Editor

PLOS ONE

Journal Requirements:

3. Thank you for submitting the above manuscript to PLOS ONE. During our internal evaluation of the manuscript, we found significant text overlap between your submission and previous work in the [introduction, conclusion, etc.].

Please revise the manuscript to rephrase the duplicated text, cite your sources, and provide details as to how the current manuscript advances on previous work. Please note that further consideration is dependent on the submission of a manuscript that addresses these concerns about the overlap in text with published work.

[If the overlap is with the authors’ own works: Moreover, upon submission, authors must confirm that the manuscript, or any related manuscript, is not currently under consideration or accepted elsewhere. If related work has been submitted to PLOS ONE or elsewhere, authors must include a copy with the submitted article. Reviewers will be asked to comment on the overlap between related submissions (http://journals.plos.org/plosone/s/submission-guidelines#loc-related-manuscripts).]

We will carefully review your manuscript upon resubmission and further consideration of the manuscript is dependent on the text overlap being addressed in full. Please ensure that your revision is thorough as failure to address the concerns to our satisfaction may result in your submission not being considered further

4. We note that you have stated that you will provide repository information for your data at acceptance. Should your manuscript be accepted for publication, we will hold it until you provide the relevant accession numbers or DOIs necessary to access your data. If you wish to make changes to your Data Availability statement, please describe these changes in your cover letter and we will update your Data Availability statement to reflect the information you provide

Reviewer's Responses to Questions

**Comments to the Author**

1. Is the manuscript technically sound, and do the data support the conclusions?

Reviewer #1: Partly

Reviewer #2: Yes

2. Has the statistical analysis been performed appropriately and rigorously? 

Reviewer #1: I Don't Know

Reviewer #2: I Don't Know

3. Have the authors made all data underlying the findings in their manuscript fully available?

Reviewer #1: No

Reviewer #2: Yes

4. Is the manuscript presented in an intelligible fashion and written in standard English?

Reviewer #1: No

Reviewer #2: No

6. PLOS authors have the option to publish the peer review history of their article (what does this mean?). If published, this will include your full peer review and any attached files.

Reviewer #1: **Yes: **Devaraj Acharya, Tribhuvan University, Nepal

Reviewer #2: No

---

## [Author Response · Author response to Decision Letter 0]

25 Jun 2023

RESPONSE TO REVIEWER

We would like to thank the reviewer for the helpful comments. Our responses to the comments from the reviewer are given beneath each comment. The added/revised text is double quoted in our response for ease of reference, with page and line numbers provided where necessary.

Reviewer#1

1. Abstract: in background: Please specify what remarkable changes have been made by Bangladesh if possible in numeric value.

Thank you for your feedback. Despite an extensive literature search, we were unable to identify any significant shifts in the barriers faced by married women of reproductive age in Bangladesh when trying to access healthcare. Furthermore, considering the word limit constraints as per the journal guideline, we were hesitant to include additional data from previous research in the abstract. Hence, no alterations were made to the abstract's background.

2. Abstract: Methods: It would be better to provide the independent variables and how did you analyze them? Also, provide the statistical software that you used for analysing the data. Did you manipulate the variables and data? If yes provide the information?

We appreciate your careful reviews and very helpful suggestions. According to your suggestions, we have added more details on the independent variables considered in the regression analyses, as follows:

“The multivariable logistic regression analysis was used, with a broad array of independent variables (such as socio-demographic factors, age, and educational level), to identify the determinants of barriers to healthcare access.” (page 2, abstract, line 15)

Given the limited word count for the abstract as per the journal guideline, we did not include information about the statistical software. Also, we did not manipulate the data.

3. Abstract: Result: Present the result of bivariate analysis also.

Thank you for your feedback. Taking into account the word limit for the abstract, we decided not to include the results of the bivariate analysis, as the findings from the multivariable regression were more robust. However, should the editors suggest the inclusion of these results, we are prepared to add them as necessary.

4. Abstract: Conclusions: Pinpoint how the policy-makers can utilize your result.

Thank you for your feedback. To make it clearer, we have edited the text as follows:

“To improve the state of women's health in Bangladesh, it is vital to consider these socio-economic factors and implement fundamental measures, such as supporting the national health policy, empowering women's socio-economic situation, and spreading the flexible way of healthcare access.” (page 2, abstract, line 29)

5. Data Availability: I would recommend writing as 'Data will be freely available after the reasonable request from the DHS program'. Since it is not possible to access the data without a request.

Thank you for your comment. To make it clearer, we have added the text as follows:

“Data will be freely available after the reasonable request from the DHS program.” (page 6, line 6)

6. Introduction: Can you present the findings of previous BDHSs? Were there any changes and which variables are remarkably changed compared to previous one? What are the possible factors for that?

Thank you for your feedback. Unfortunately, there were no prior studies or reports from the BDHS that focused on healthcare access in Bangladesh. Consequently, it was not feasible to make comparisons with past BDHS results. 

7. Materials and methods: More elaboration is needed here. Follow STROBE guidelines. I am confusing the sentence: 'In the second stage, 20,250 households were selected for data collection,….' Please revisit it.

Thank you for your feedback. To make it clearer, we have edited the text as follows:

“In the second stage, a systematic sample of an average of 30 households per EA was selected for urban and rural areas separately, and for each of the eight divisions. Based on this design, 20,250 residential households were selected.” (page 6, line 23)

8. Statistical analysis: Citation required here at: ..' STATA/MP version 17.0 was used for analysis.' How did you assess the variables in multivariate analysis? Did you assess the multi-collinearity issue? What is the basis of assessing the variables?

Thank you for your comment. Thank you for your comment. To clarify, we assessed the issue of multicollinearity by examining correlation coefficients. For greater clarity, we have added the text as follows:

“Thirdly, we assessed the multi-collinearity issue by verifying the correlation coefficients. Referring to the assessment, we included all the independent variables in the logistic regression model.” (page7, line33)

9. Statistical analysis: In the multivariate analysis, there is no consistency in assuming reference value? For example, last attribute/category in age group, while first category/attributes for others? What is the cause of that? Sometimes, it will be interesting if you make the middle attribute as reference category such as in wealth quintile.

Thank you for your significant comment. The establishment of the reference group was determined taking into account the reference groups from prior studies using the DHS in other countries, as well as the ease of interpreting and discussing the regression results.

10. Result: Present the impression of the table rather than an explanation of the table exactly. What is the basis of 5-year age category?

Thank you for your comment. We have thoroughly revised the description for Table 1 as follows:

“Table 1 presents the socio-demographic characteristics of the participants. Approximately half of the participants (45.8%) were under 30 years old, and the majority were married (94.3%). The majority had completed secondary school (87.5%), and agriculture was the most common occupation (32.7%), while nearly half were unemployed (49.8%). Few participants reported regularly listening to the radio or reading newspapers (with 95.2% and 90.7% respectively stating they did so 'not at all'), however, over half (55.0%) stated they watched television at least once a week. The prevalence of mobile phone ownership was 60.2%, and 71.5% of respondents resided in urban areas. (Table 1 about here)” (page 9, line 3)

As with the response to comment 9, the establishment of the 5-year age group was determined taking into account prior studies using the DHS and the ease of interpretation.

11. Result: … 5.7% of respondents were widowed, divorced..' Do not start the sentence with number. You may write as 'almost six (5.7) percent of the total respondents…'

In table 3: Wealth Index: The reference category in the middle would be interesting to compare with the poorest and the richest.

Thank you for your comment. We have restructured the description for Table 1, including the text you pointed out. Please refer to our response to comment 10.

12. Result: In table 3: Residence: Rural woman OR = 0.77. It is interesting. Are the rural women rich in Bangladesh? ….The women who lived in rural areas had less likely to have financial barrier compared to those women who lived in urban areas...Am I right? Please present the possible causes for that.

Thank you for your comment. To make it clearer, we have added the text in the discussion section as follows:

“A women’s place of residence is also an important determinant of the barriers they may face in accessing healthcare. Our study indicates that women living in rural areas are less likely to face financial barriers. Bangladesh is one of the countries where increasing income disparity is of concern with a Gini coefficient of 0.458 (2010) [49]. Furthermore, the income Gini coefficients in rural and urban areas were 0.431 (2010) and 0.452 (2010), respectively [49]. This suggests that income inequality tends to be more severe in urban areas than in rural areas. Therefore, income disparity might be one factor making it harder for urban women, compared to those in rural areas, to secure the necessary financial resources for healthcare.” (page 13, line 32)

13. Result in table 3: It is not necessary to indicate ** for significant value. The value of 95%CI shows whether it is significant or not. Present the model summary of each dependent variables. It shows the predictive capacity of the table. If possible provide value of AUR (Area Under Curve).

Thank you for your feedback. To allow readers to easily determine the significance of the estimates at a glance, we have kept the p-values and asterisks. Additionally, we have incorporated the AUR into Table 3. To be able to check it easier, I have attached Table 3 on page 8 of this file.

14. Discussion: Present your discussion theme wise i.e. Autonomy, Money, Proximity, Alone etc. Add these themes as sub heading of the discussion section.

Thank you for your comment. After careful re-discussion with authors experienced in policy formulation and strategic proposals, we concluded that the variable-wise discussion provide a more practical and useful entry point for policymakers. Therefore, we opted to structure our discussion based on variables rather than barriers.

To make it clearer, we have edited the text as follows:

“From the following, this study focused on the main variable we identified as independent variables to discuss useful suggestions for policy making.” (page12, line11)

15. Discussion: ..'Our study identified that the proportion of the barriers to access to healthcare in Bangladesh…' Present the exact value.

Thank you for your comment. To make it clearer, we have edited the text as follows:

“Our study identified that the proportion of the barriers to access to healthcare in Bangladesh (66.3%) is higher than other reports…” (page12, line14)

16. Conclusion: Please be specific while suggesting to the policy-makers. How your findings can be translated into policy-making?

Thank you for your comment. In order not to overstate what can be scientifically inferred from the research results, we have made the following additions:

“To improve the state of women's health in Bangladesh, it is vital to consider these socio-economic factors and implement fundamental measures, such as supporting the national health policy, empowering women's socio-economic situation, and spreading the flexible way of healthcare access.” (page14, line25)

17. Ethical Issue: I think, ethical approval was taken by organization while conducting BDHS.

Thank you for your comment. Indeed, the BDHS procedures and questionnaires are ethically approved by the ICF Institutional Review Board (IRB). However, the journal guidelines instruct us to detail information about the ethical review of our study in the "Ethical approval" section. In this study, as we are using open data from the BDHS, an ethical review was not necessary. Nevertheless, given the importance of your comment, we have added information about the BDHS IRB in the method section where we explain the BDHS, as follows: 

“The survey protocol was approved by institutional review boards (IRBs) at ICF and the Bangladesh Medical Research Council (BMRC). Both IRBs and BMRC approved the protocols before the commencement of data collection activities.” (page6, line7)

18. Ethical approval and consent to participate: I would recommend to write as '…. The survey protocol was approved by institutional review boards (IRBs) at ICF and the Bangladesh Medical Research Council (BMRC). Both IRBs and BMRC approved the protocols before the commencement of data collection activities. Consent was taken before administered…. '

Thank you for your input. As with our response to comment 17, we have not been directly involved with the BDHS itself; we merely utilized their publicly available data. Therefore, we have included details about the BDHS IRB in the methods section. If the editors instruct that the ethical approval of BDHS is needed in this "Ethical approval" footnote, we plan to move the description here.

19. Data availability: Revise your sentence. It is not possible to access the data without login and proposal. There is no data accessible through the link.

Thank you for your comment. We have made the necessary corrections.

20. Others: Merge the citation as required. Provide links or DOI as possible and applicable so that readers could visit the original sources easily. Revisit the paper for grammar and sentence structure. Somewhere, there is no consistency within and among the sentences and paragraphs.

Thank you for your valuable comment. In line with the journal guidelines, we have edited the reference list. As the journal guidelines do not include links or DOIs in the citation format, we have chosen not to include them.

Reviewer#2

General comment: 

1. The authors have worked on the document and expressed their issues with proper objectives. There are no issues to publish this paper. However, there are some concerns in the paper. The transition of the introduction is not well. The author must try to write it smoothly and transition the global to the scenario of Bangladesh in a smooth fashion. Furthermore, I am not well convinced on the way why this research is required, therefore, please extend your introduction part so that you can grab the attention of the readers.

“Globally, women experience healthcare inequalities, which may contribute to excessive mortality rates at various stages of their lives [6]. Reports have highlighted that, in addition to education levels and poverty, numerous social, cultural, and geographical factors are associated with poor utilization and access to healthcare services [7] [8] [9]. Furthermore, previous research has indicated that individual and household factors, such as marital status [10], educational attainment [11], and wealth index [12], may be linked to women's access to healthcare services. For instance, despite the majority of maternal deaths being considered preventable, approximately 295,000 women worldwide died during pregnancy or within the postpartum period in 2017. 

On a global scale, Bangladesh is among the countries that have made significant progress in reducing maternal and child mortality rates [13]. Maternal mortality has decreased from 297 deaths per 100,000 live births in 2007 to 173 deaths per 100,000 live births in 2017 [4], and under-5 mortality has decreased from 49 deaths per 1,000 live births in 2010 to 29 deaths per 1,000 live births in 2020 [14]. The success of Bangladesh's "Maternal Health Strategy 2001" and its subsequent successor, the "Bangladesh National Strategy on Maternal Health (BNSMH) 2017-2030," likely underlie this progress. These strategies aim to address existing disparities and inequities in the provision of quality maternal healthcare services and to tackle the social and developmental factors that affect maternal health [13]. Despite the remarkable progress in healthcare, particularly in maternal and child health outcomes, Bangladesh is still considered to be in the process of improving healthcare access, especially ensuring access to primary and emergency healthcare services for all. Various discussions have been held regarding barriers to access, such as inadequate financial resources [17], a shortage of skilled personnel [18], and economic disparities [17], but scientific evidence regarding the scale and factors contributing to these access barriers is extremely limited. 

The objective of this study is to examine the determinants and barriers to healthcare access among women of reproductive age in Bangladesh, using nationally representative data. The insights gained from this study are expected to be useful for healthcare policymakers in achieving healthcare equity, improving women's healthcare through the redistribution of health services in Bangladesh, and further reducing maternal and child mortality rates.” (page 4, line 13)

Major Comment:

1. Page no. 3, Line 10: LMICs- Low and Middle Income Countries (you have written Low and Lower and Middle Income Countries). 

Thank you for your feedback. we have made the necessary revisions to the text.

2. Page 5: Data Source- It would be wise to mention the ethical clearance issue of the data. 

Thank you for your feedback. We have incorporated the ethical considerations into the "Ethical approval" section of materials and methods as follows:

"As the de-identified data for the current study came from secondary sources whose data is publicly available, ethics approval for this study was not required. This study was conducted according to the guidelines in the Declaration of Helsinki." (page 6, line 11)

Regarding the BHDS itself, although we are not directly involved in it, another reviewer suggested including information about its ethics. Therefore, we have added the following statement to the methods section: 

"Data will be freely available after a reasonable request from the DHS program. The survey protocol was approved by institutional review boards (IRBs) at ICF and the Bangladesh Medical Research Council (BMRC). Both IRBs and BMRC approved the protocols before the commencement of data collection activities." (page 6, line 7)

3. Page 5, Line 15: Please check the numbers. 

Thank you for your comment. we have made the necessary revisions to the text. 

Table 3: Multivariable logistic regression of individual and community level factors associated with barriers to healthcare access among ever-married women aged 15-49 in Bangladesh, 2017-18 (n=20,127)

Characteristics Permission Money Distance Alone At least one barrier

 AOR [95% CI] AOR [95% CI] AOR [95% CI] AOR [95% CI] AOR [95% CI]

Age 

15-19 1.34 [1.08-1.66]** 0.69 [0.60-0.81]*** 1.06 [0.92-1.23] 1.54 [1.34-1.78]*** 1.17 [1.00-1.37]

20-24 1.09 [0.90-1.33] 0.84 [0.73-0.96]** 1.13 [0.99-1.28] 1.17 [1.03-1.33]* 0.99 [0.86-1.13]

25-29 1.05 [0.87-1.28] 0.94 [0.83-1.07] 1.10 [0.97-1.25] 1.07 [0.95-1.21] 1.02 [0.89-1.17]

30-34 0.93 [0.77-1.12] 1.06 [0.94-1.20] 1.14 [1.00-1.28]* 0.99 [0.87-1.11] 1.01 [0.88-1.15]

35-39 0.95 [0.79-1.14] 1.03 [0.91-1.17] 1.08 [0.95-1.22] 0.95 [0.84-1.07] 0.97 [0.85-1.11]

40-44 0.79 [0.65-0.96]* 0.95 [0.83-1.08] 0.98 [0.86-1.12] 0.98 [0.87-1.11] 0.94 [0.82-1.08]

45-49 Ref Ref Ref Ref Ref

Marital status 

Married - Ref - - Ref

Widowed - 1.53 [1.26-1.84]*** - - 1.13 [0.92-1.38]

Divorced - 1.91 [1.47-2.48]*** - - 1.15 [0.88-1.51]

No longer living together/separated - 1.98 [1.46-2.69]*** - - 1.22 [0.88-1.69]

Educational level 

No education Ref Ref Ref Ref Ref

Primary 0.89 [0.77-1.03] 0.83 [0.75-0.92]*** 0.84 [0.76-0.93]*** 0.94 [0.86-1.04] 0.87 [0.78-0.97]*

Secondary 0.81 [0.68-0.95]** 0.61 [0.55-0.69]*** 0.75 [0.68-0.84]*** 0.84 [0.76-0.93]** 0.68 [0.60-0.77]***

Higher 0.61 [0.47-0.78]*** 0.39 [0.33-0.46]*** 0.61 [0.52-0.71]*** 0.63 [0.55-0.73]*** 0.49 [0.41-0.57]***

Occupation 

Not working Ref Ref Ref Ref Ref

Professional/technical/managerial 0.71 [0.43-1.18] 0.57 [0.40-0.80]** 0.75 [0.57-0.99]* 0.65 [0.49-0.85]** 0.71 [0.55-0.91]**

Sales 1.25 [0.85-1.82] 1.20 [0.92-1.55] 1.08 [0.84-1.40] 0.75 [0.58-0.96]* 1.02 [0.79-1.32]

Agricultural 0.92 [0.81-1.06] 1.16 [1.06-1.26]** 1.08 [0.99-1.17] 1.03 [0.94-1.11] 1.20 [1.09-1.31]***

Household and

domestic services 0.93 [0.63-1.36] 2.07 [1.59-2.69]*** 1.22 [0.95-1.55] 0.99 [0.78-1.25] 1.31 [1.00-1.73]

Services 0.85 [0.65-1.13] 1.28 [1.08-1.52]** 0.91 [0.77-1.08] 0.74 [0.63-0.88]*** 0.88 [0.74-1.05]

Manual 0.76 [0.62-0.94]* 1.26 [1.11-1.42]*** 0.93 [0.82-1.05] 0.88 [0.78-0.99]* 1.08 [0.95-1.22]

Religion 

Islam - Ref Ref Ref Ref

Hinduism - 1.08 [0.94-1.25] 1.06 [0.93-1.22] 1.16 [1.02-1.33]* 1.11 [0.96-1.29]

Other (Buddhism &Christianity) - 1.56 [0.98-2.50] 1.55 [0.99-2.40] 1.40 [0.92-2.15] 1.51 [0.91-2.52]

Covered by health insurance 

No - - - - -

Yes - - - - -

Frequency of listening to radio 

Not at all - Ref Ref Ref Ref

Less than once a week - 0.85 [0.69-1.06] 0.95 [0.77-1.15] 1.12 [0.93-1.36] 0.96 [0.79-1.17]

At least once a week - 0.83 [0.64-1.06] 0.88 [0.70-1.11] 0.99 [0.79-1.23] 0.95 [0.76-1.19]

Frequency of reading newspaper or magazine 

Not at all Ref Ref Ref Ref Ref

Less than once a week 0.84 [0.66-1.07] 0.69 [0.58-0.81]*** 0.83 [0.72-0.96]* 0.75 [0.65-0.87]*** 0.72 [0.62-0.82]***

At least once a week 0.72 [0.49-1.08] 0.67 [0.52-0.86]** 0.89 [0.72-1.11] 0.67 [0.54-0.83]*** 0.65 [0.53-0.79]***

Frequency of watching television 

Not at all Ref Ref Ref Ref Ref

Less than once a week 1.15 [0.97-1.36] 1.03 [0.91-1.16] 0.88 [0.78-0.98]* 0.80 [0.71-0.89]*** 0.98 [0.86-1.12]

At least once a week 0.87 [0.77-0.99]* 0.93 [0.85-1.01] 0.89 [0.82-0.97]** 0.83 [0.77-0.90]*** 0.88 [0.80-0.96]**

Owns a mobile telephone 

No Ref Ref Ref Ref Ref

Yes 0.81 [0.73-0.90]*** 0.79 [0.73-0.84]*** 0.91 [0.85-0.97]** 0.85 [0.79-0.91]*** 0.78 [0.73-0.84]***

Sex of household head 

Male - - - Ref -

Female - - - 0.88 [0.81-0.96]** -

Wealth index 

Poorest Ref Ref Ref Ref Ref

Poorer 0.84 [0.72-0.98]* 0.67 [0.60-0.74]*** 0.82 [0.74-0.91]*** 0.97 [0.88-1.08] 0.85 [0.75-0.96]*

Middle 0.91 [0.77-1.08] 0.45 [0.40-0.51]*** 0.75 [0.67-0.84]*** 1.01 [0.91-1.13] 0.67 [0.59-0.76]***

Richer 0.84 [0.70-1.02] 0.32 [0.28-0.36]*** 0.68 [0.60-0.77]*** 1.05 [0.93-1.18] 0.60 [0.53-0.69]***

Richest 0.73 [0.59-0.92]** 0.20 [0.17-0.24]*** 0.55 [0.48-0.64]*** 0.87 [0.76-1.00] 0.45 [0.38-0.52]***

Residence 

Urban Ref Ref Ref Ref Ref

Rural 1.32 [1.02-1.72]* 0.77 [0.67-0.89]*** 1.49 [1.29-1.72]*** 1.27 [1.12-1.44]*** 1.18 [1.02-1.36]*

Region 

Dhaka Ref Ref Ref Ref Ref

Barisal 0.75 [0.49-1.16] 0.79 [0.62-1.01] 1.19 [0.93-1.53] 1.12 [0.90-1.40] 0.92 [0.71-1.20]

Chittagong 0.93 [0.67-1.29] 0.84 [0.69-1.03] 0.86 [0.71-1.05] 0.93 [0.78-1.11] 0.78 [0.64-0.95]*

Khulna 0.95 [0.66-1.38] 1.03 [0.83-1.28] 1.03 [0.83-1.27] 1.17 [0.97-1.42] 1.02 [0.82-1.26]

Mymensingh 0.53 [0.35-0.81]** 0.94 [0.75-1.19] 0.79 [0.63-1.00] 0.90 [0.73-1.11] 0.78 [0.61-0.99]*

Rajshahi 0.92 [0.64-1.31] 0.66 [0.53-0.81]*** 0.75 [0.60-0.92]** 0.83 [0.69-1.00]* 0.70 [0.56-0.86]**

Rangpur 0.97 [0.67-1.40] 0.86 [0.69-1.07] 0.92 [0.74-1.14] 0.97 [0.80-1.18] 0.90 [0.71-1.13]

Sylhet 0.88 [0.58-1.35] 0.77 [0.60-0.98]* 0.97 [0.76-1.24] 1.00 [0.81-1.25] 0.74 [0.58-0.96]*

Not dejure resident 1.01 [0.77-1.31] 0.77 [0.65-0.91]** 0.80 [0.67-0.94]** 0.88 [0.75-1.02] 0.64 [0.54-0.76]***

AUC 0.83 [0.82-0.83] 0.77 [0.77-0.78] 0.74 [0.73-0.74] 0.71 [0.70-0.71] 0.76 [0.75-0.76]

*p<0.05, **p<0.01, ***p<0.001

AOR: adjusted odds ratio, CI: confidence interval, Ref: reference, AUR: Area under curve

---

## [Editor Report · Decision Letter 1]

17 Jul 2023

Factors associated with barriers to healthcare access among ever-married women of reproductive age in Bangladesh: analysis from the 2017-2018 Bangladesh Demographic and Health Survey

PONE-D-23-06003R1

Dear Dr. Hitomi,

We’re pleased to inform you that your manuscript has been judged scientifically suitable for publication and will be formally accepted for publication once it meets all outstanding technical requirements.

Kind regards,

Ramesh Adhikari

Academic Editor

PLOS ONE
---

## [Editor Report · Acceptance letter]

27 Jul 2023

PONE-D-23-06003R1 

Factors associated with barriers to healthcare access among ever-married women of reproductive age in Bangladesh: analysis from the 2017-2018 Bangladesh Demographic and Health Survey 

Dear Dr. Hinata:

I'm pleased to inform you that your manuscript has been deemed suitable for publication in PLOS ONE. Congratulations! Your manuscript is now with our production department. 

Kind regards, 

on behalf of

Professor Ramesh Adhikari 

Academic Editor

PLOS ONE